# Development of the macaque face-patch system

Margaret S. Livingstone[1,*], Justin L. Vincent[1,*], Michael J. Arcaro[1,*], Krishna Srihasam[1], Peter F. Schade[1] & Tristram Savage[1,†]

Face recognition is highly proficient in humans and other social primates; it emerges in infancy, but the development of the neural mechanisms supporting this behaviour is largely unknown. We use blood-volume functional MRI to monitor longitudinally the responsiveness to faces, scrambled faces, and objects in macaque inferotemporal cortex (IT) from 1 month to 2 years of age. During this time selective responsiveness to monkey faces emerges. Some functional organization is present at 1 month; face-selective patches emerge over the first year of development, and are remarkably stable once they emerge. Face selectivity is refined by a decreasing responsiveness to non-face stimuli.

[1] Department of Neurobiology, Harvard Medical School, Boston, Massachusetts 02115, USA. * These authors contributed equally to this work. † Present address: Department of Psychology, University of Washington, Seattle, Washington 98104, USA. Correspondence and requests for materials should be addressed to M.S.L. (email: mlivingstone@hms.harvard.edu).

In adult humans and macaques, inferotemporal cortex (IT) is subdivided into functional domains that are selective for different image categories and are critical for high-level object recognition, but it is unknown how these domains develop[1–3]. The biological importance of several category domains, such as faces, bodies and scenes[1,2,4], and their stereotyped locations across individuals, in both humans and monkeys, suggest a role for innate mechanisms[3]. However, human IT also has domains selectively responsive to categories of man-made things, like buildings and text[5]; the existence of domains for such unnatural categories argues for a role for experience in their development. Further evidence for the importance of experience in IT domain formation is our previous finding that intensive early experience can cause the formation of domains in stereotyped locations in monkey IT for unnatural object categories that monkeys never normally experience[6,7]. How innate programs interact with experience to form highly selective domains is unclear, as is the normal time course of their development[3,8,9].

Here we monitored the emergence and refinement of the most well-studied IT domain, face patches, longitudinally from 1 month to more than 2 years of age using functional MRI (fMRI) of alert macaques over their normal development. We contrasted responses to static images of faces with responses to images of rectilinear objects and found that face patches emerged around 200 days of age, and were remarkably stable after that. In the youngest, least attentive, animals we used movie clips that revealed some selectivity for faces compared to scenes as early as 1 month of age. To figure out whether the very early selectivity to the movies represented face selectivity *per se* or some low-level selectivity to shapes that include faces, we compared responses to faces and objects to responses to scrambled faces, which share local image features with faces, and we compared the selectivity of the responses in IT to responses in V1 and in subcortical structures. We determined that the very early face selectivity was weaker than the selectivity that emerged by 200 days of age. The early broad selectivity for faces was in the same location as the later, stronger, selectivity, indicating the existence of an early proto-organization. Face selectivity emerged primarily via a decrement in responsiveness to non-face images, indicating a role for pruning in generating category-selective domains. These results indicate an early, probably innate, proto-organization of IT that is sculpted by experience.

## Results

**Emergence of face domains.** We scanned four infant macaque monkeys (three male, one female) longitudinally over their development. They were housed with their mothers in a room with other monkeys for the first 4–6 months, then co-housed with other juveniles. Thus, they all had intensive early experience with faces. For scanning they were alert, and their heads were immobilized non-invasively using a foam-padded helmet with a bite bar. The longitudinal development of monkeys B1 and B2 was coarsely sampled, whereas monkeys B3 and B4 were scanned extensively from 10 days to more than 2 years of age. The visual stimuli are shown in Fig. 1 and consisted of either blocks of static images (monkey faces, scrambled monkey faces, rectilinear objects) or video clips (monkeys grooming each other with prominent faces, the same monkey videos but with the faces pixelated out, inanimate dynamic scenes); the movies were used only at the youngest time points.

We used a monocrystalline-iron-oxide-nanoparticle contrast agent (MION; Feraheme) that increases signal-to-noise relative to the BOLD signal, and allows measurement of blood volume directly[10]. We reliably obtained clear visual response signals in V1 for single scans, allowing us to use the V1 signal as a quality control; we eliminated blocks or entire scans when the monkey was inattentive or not looking at the stimulus (the number of accepted blocks for each image category for each scan session is listed in Supplementary Table 1).

Each monkey exhibited a distinct pattern of faces > objects activations (Fig. 2) that corresponded to the previously described macaque face patches in IT[11]. In contrast with previous studies we did not reproducibly see any face selectivity at the anteriormost tip of the superior temporal sulcus (STS; patch AM) or in the frontal lobe[12]; we cannot tell whether this is because our young animals do not work as long as the adult animals in which these patches were described, or because the development of these regions is relatively delayed. Once these face-selective regions appeared, they were present in the same location in subsequent sessions, even down to minor stereotyped differences between hemispheres or between monkeys. A pattern of face selectivity (red patches) that was virtually identical to the pattern at 2 years of age (in the same monkey) was clear by 277 days of age in monkey B1, 153 days in monkey B2, 207 days in monkey B3 and 199 days in monkey B4.

The regions responsive to objects > faces (blue in Fig. 2) were more variable, but lay between the face patches within the STS as well as on the inferior bank of the STS. It is apparent that early visual areas (see Supplementary Fig. 1 for map of visual areas) were usually more activated by objects than by faces; analysis of images revealed that the object stimuli were indeed more variable in colour and in retinotopic distribution and were thus probably stronger stimuli for early visual areas than were the (more homogeneous within-category) face stimuli. This is a benefit since we can therefore conclude that the face patches we observed in these sessions were genuinely more responsive to faces than to objects, rather than reflecting any differential retinotopic stimulation due, for example, to their inherent greater interest to the monkeys.

Once established, the unique pattern of faces > objects patches in each monkey, in each hemisphere, was remarkably stable. This same pattern was apparent as weak faces > objects activations prior to 200 days of age in monkeys B1, B3 and B4, and is apparent at early ages (81, 88, 102 days old) in monkey B3 as holes (that is, lack of objects > faces activations) in the object-selective activations (blue) at the same loci as the face patches at older ages. (Note that where the age is blue in Fig. 2, there were no significant faces > object activations at all in the STS, so the threshold was lowered to $P < 0.25$ to reveal any statistically insignificant organization that might be present). Thus, results using static images indicate a stable organization for face selectivity that is robust and stable by 200 days of age. But these maps also suggest the presence of an organization that was present, and similar in spatial organization, even earlier. We quantified this similarity in monkeys B3 and B4 (for whom we have extensive longitudinal data) by correlating the spatial pattern of faces-minus-objects activations between each session and every other session in an anatomically defined region of interest (ROI) consisting of the lower bank and lip of the STS. The mean pairwise correlations reached a stable plateau by 200 days (Supplementary Fig. 3), but were significantly above zero for all the static-image sessions. This impression of an organization that precedes significant face-selective domains is supported by data collected at even earlier ages using movie stimuli.

**Very early organization revealed by movie data.** We scanned these monkeys at even earlier ages using movie stimuli. We used movies for the youngest monkeys because infants < 2 months old did not look at the screen during static-image presentation but would look, if erratically, at movies. The movies consisted

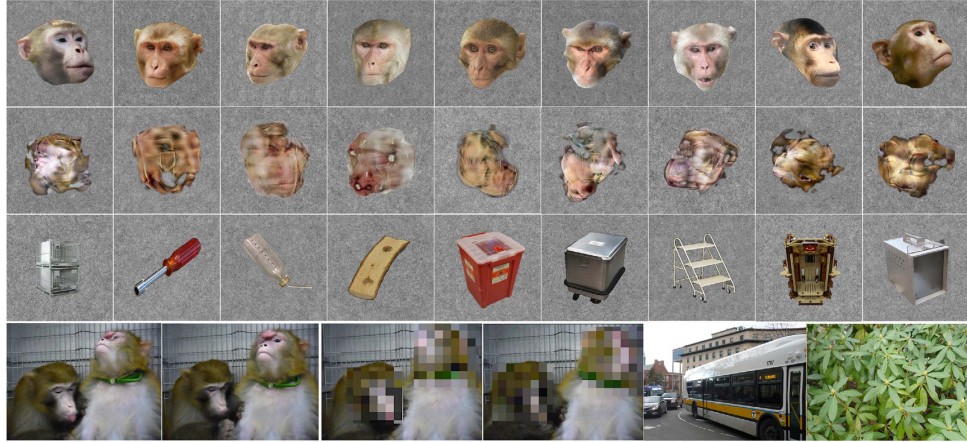

**Figure 1 | Visual stimuli used for longitudinal scanning.** The first three rows show a subset of the static images that were shown in blocks; the bottom row shows examples of the videos that were shown to the monkeys younger than 2 months. (Top row) Monkey faces. (Row 2) Scrambled monkey faces, scrambled using the method of Portilla and Simoncelli[51]. (Row 3) Familiar rectilinear objects. (Row 4) The three categories of videos shown to monkeys <2 months old: L to R: monkeys grooming each other with prominent faces; same movies with the faces pixelated; traffic or time-lapse nature.

of 30 second video clips of (1) monkeys moving around and grooming each other with large faces present at all times, (2) the same movie clips but with the faces pixelated out, and (3) dynamic scenes of traffic or time-lapse nature scenes. In older monkeys, the contrast faces-minus-scenes movies activated the same regions as did the faces-minus-objects static-image contrast (Supplementary Fig. 4a,b), justifying the use of these movies in the youngest monkeys. All three monkeys tested with movie stimuli at very young ages showed some selectivity in the STS for faces>scenes in the movie data prior to 200 days of age, even as early as 1 month of age (Fig. 3a; Supplementary Fig. 4c), though no functional organization was found at ages younger than 1 month (Supplementary Fig. 4c). The location of the faces>scenes movie selectivity in the younger monkeys was in roughly the same location as the faces>objects static-image activations at older ages in the same monkeys (cf. white outlines in Fig. 3 and Supplementary Fig. 4c), suggesting the existence of some functional organization as early as we could detect visual responsiveness in IT.

Although the movie activations indicated some organization in IT at very early ages, the selectivity for face movies extended to some non-face stimuli, and there were differences between the month-old monkeys and the older monkeys, indicating instead that the very early selectivity was likely not categorical. This is apparent by the fact that pixelated-face>scenes movies activations were in the same locations as the faces>scenes movies (Fig. 3b; Supplementary Fig. 4c), and by the lack of significant faces>pixelated movies in the younger monkeys. In older monkeys, the pixelated-face>scene activations were also in the same locations as the faces>scenes movies. To compare the selectivity of the face patches in the month-old monkeys to older monkeys, in Fig. 3c we used the anterior and posterior face patches, calculated from different scan sessions at older ages using static-image data, as ROIs to measure the activations in response to face movies (red), pixelated-face movies (green) and inanimate scene movies (blue) for (top) 3 monkeys before the static-image face patches appeared (B2 at 48 days, B3 at 38 days and B4 at 30 days), and (bottom) for two older monkeys after the static-image face patches had appeared (B1 at 180 days and B4 at 836 days). In both the younger and older monkeys, these ROI responses to face movies were significantly larger than to scene movies in the face patches (young monkeys $t(11) = 8.39$, $P = 4.1 \times 10^{-6}$; older $t(7) = 11.79$, $P = 7.17 \times 10^{-6}$, 2-tailed

$t$-test). To increase statistical power, comparisons were run on the combined face patches, but comparable results were observed for each ROI separately. In the younger monkeys, the ROI responses to face movies were not significantly larger than responses to pixelated-face movies, whereas in the older monkeys they were (young monkeys $t(11) = 1.55$, $P = 0.15$; older $t(7) = 8.62$, $P = 5.6 \times 10^{-5}$, 2-tailed $t$-test), indicating that the faces>scenes activations in the youngest monkeys were not due to selectivity for faces *per se*. Although there are a small number of data points for these significance tests, the results are clear in the maps from all the individual subjects (Fig. 3; Supplementary Fig. 4). Thus, these early movie data support the existence of a very early functional organization that precedes face-patch formation, that is, a proto-organization.

Obviously, the dynamic movies are very different from blocks of static images, and include other category-distinguishing features such as retinotopic diversity, biological motion, bodies and other motion features. Previous studies in humans and macaques have shown that dynamic faces activate regions in addition to those activated by static faces[13,14]. Even though we have concerns about the movie data because of these confounds, as well as the small number of acceptable scans acquired during these sessions, we include these data in our analyses because they represent the earliest data we could obtain. Throughout this manuscript we distinguish the movie data from the later static-image data.

**Category versus low-level features.** Is the failure to find robust category selectivity prior to 200 days because selectivity had not yet emerged, because the monkeys did not look at, or see, the images reliably, because the vascular response to neuronal activity was immature[15,16], or because neuronal responses in IT were weak[17–19]? To see if there was any responsiveness or selectivity before 200 days in what would become face patches, we first measured the raw signal time courses. For quantitative ROI analyses, we defined a V1 ROI (covering the central 6–7° of visual field using retinotopic mapping at >2 years of age), and used the data from monkey B3 on day 284 and from monkey B4 on day 276 to calculate face and object-selective ROIs (see Fig. 2, top centre). We combined AL and AF into a single bilateral anterior-face ROI, and ML and MF into a single bilateral middle-face ROI; the object ROI was defined as object-selective

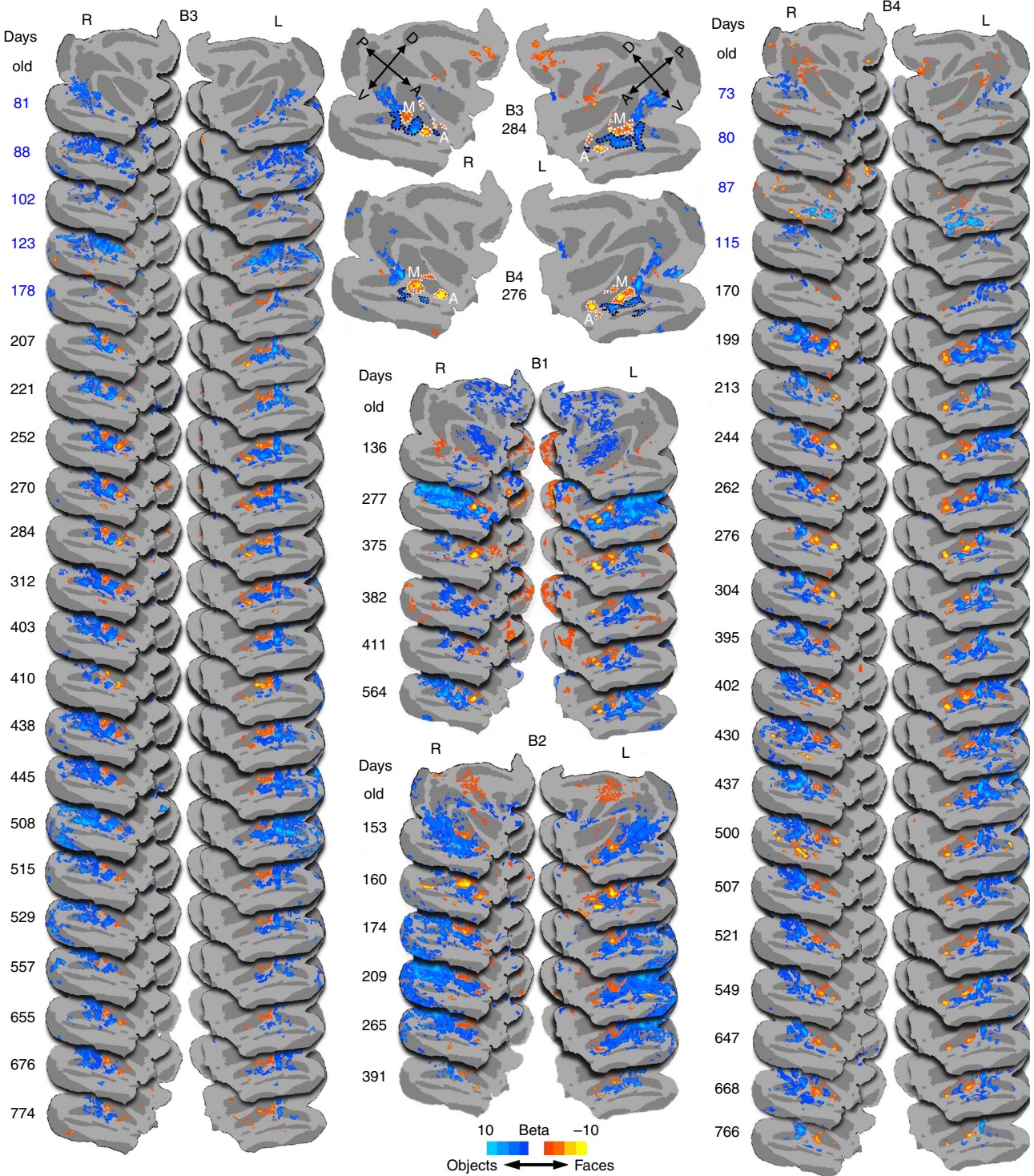

**Figure 2 | Development of face selectivity.** Maps of activations (beta coefficients) for the contrast faces-minus-objects for each static-image scanning session for each of four young monkeys projected onto the standard F99 macaque brain flat map[57,58]. Each monkey's age in days is indicated beside each map. Where the age is indicated in black, the beta coefficients were thresholded at $P \leq 0.05$, FDR corrected; where the age is indicated in blue, no faces > object activations were present in the STS at $P < 0.05$, so the beta coefficients were thresholded at $P \leq 0.25$, FDR corrected. In the top centre are enlarged maps for monkeys B3 and B4 for the dates we used for defining the ROIs for quantitative ROI analysis; M indicates the region outlined in white corresponding to the middle face-patch cluster, and A indicates the region outlined in white corresponding to the anterior face-patch cluster; the object patches are outlined in black. Corresponding visual areas are mapped in Supplementary Fig. 1 and non-overlapping maps are shown in Supplementary Fig. 2.

cortex anterior to V4. These ROIs were projected onto data from other scan sessions for the same monkey. Figure 4 shows the average signal time courses in response to faces/face movies (red), objects/scene movies (blue) and scrambled faces/pixelated-

face movies (green) for 1 movie session, and 3 static-image sessions before the face patches became stably significant compared to six dates after. These traces show, first, that the responses in an ROI for central V1 were significant and robust at

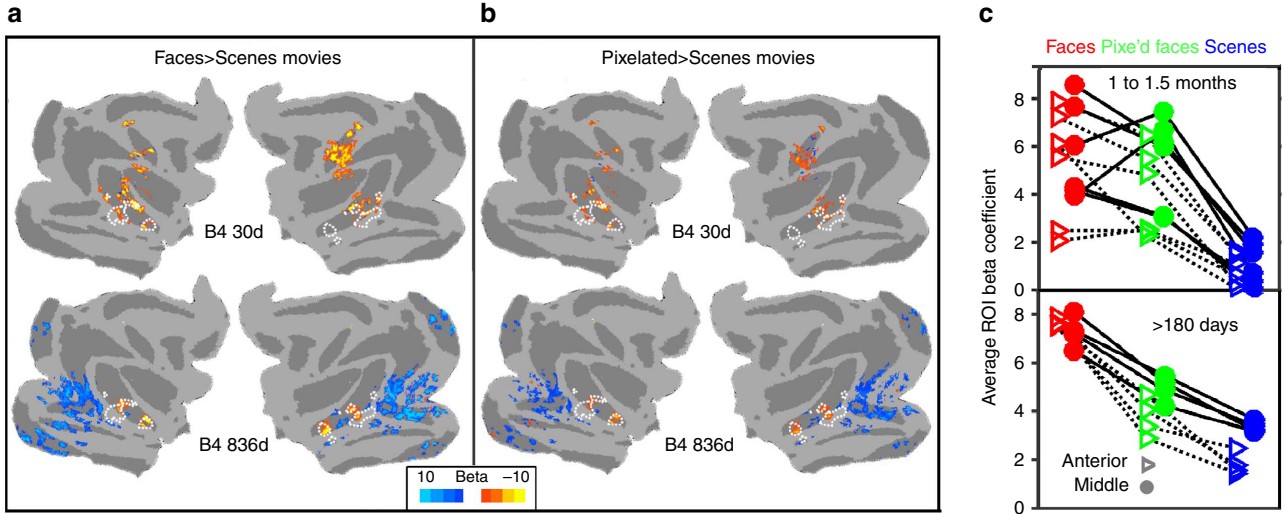

**Figure 3 | Very early organization revealed using movies.** (**a**) Representative activations for the contrast faces-minus-scenes movies in one monkey at two ages, once before static-image face patches appeared at 30 days old (above) and once after the static-image face patches appeared, 836 days (below), both thresholded at $P < 0.05$ FDR corrected. Dotted white outlines indicate faces > object static-image activations for the same monkey at 276 days old. Additional examples of movie data for other young monkeys are shown in Supplementary Fig. 4. (**b**) Maps of the contrast pixelated faces > scenes movies for the same monkey. (**c**) ROI analysis of three infant (top) and two juvenile (bottom) monkeys (two hemispheres each) in response to movies. Activations in anterior (triangles) and middle (circles) face patches in response to movie stimuli for (top) three monkeys at the youngest ages at which they showed significant faces > scenes movie activations and (bottom) two monkeys at ages after the static-image face patches had appeared.

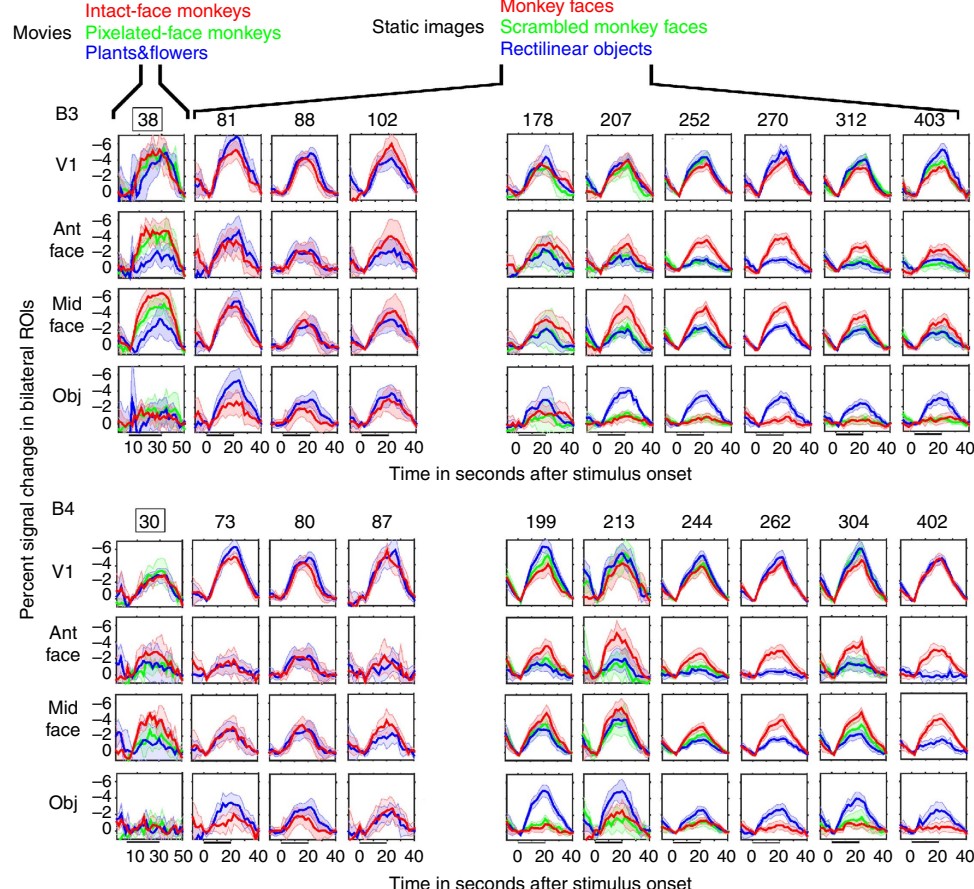

**Figure 4 | Average raw response time courses.** Each panel shows the normalized MR signal time course in the ROI indicated on the left, averaged over blocks for each stimulus category; shading indicates 95% confidence limits. Face and object ROIs were identified using data from monkey B3 at 284 days old and from monkey B4 at 276 days old. The central V1 ROI (central 6–7° of visual field) was identified by retinotopic mapping in each animal at > 2 years of age. Numerals above each column indicate the monkey's age in days; outlined age indicates movie stimuli; others were blocks of static images.

every session. This validates that, for each block included in the averages, the monkeys were indeed looking at the stimuli, and the cortex did show strong vascular responses to visual stimuli compared to grey background.

The first column in Fig. 4 shows the response time courses from the second movie session, when both monkeys showed significant faces > scene movie activations in the same locations that would become face selective at older ages. Even though the responses in the future face patches to face movies were indeed larger than responses to scene movies, there is large variance across repetitions, and overlap in the confidence limits for face movies compared to pixelated movies. Similarly, the responses for 73 to 102 days old show strong responses in all ROIs to both faces and objects, but no differential responsiveness. Thus, the lack of selectivity at these early ages was not due to a lack of responsiveness to what would become the preferred stimulus, but rather to a lack of differential responsiveness to what would become preferred and non-preferred stimuli.

In contrast to the data from the first few months of age, after day 178 for monkey B3 and day 199 for monkey B4, the face-patch ROIs showed a consistently large response to faces but diminishing (across development) responses to objects or scrambled faces. The response pattern was less clear in the object patches, but we don't know what the actual preferred stimuli should be for those regions. Given that selectivity must arise either by increasing responsiveness to preferred stimuli[20] or by decreasing responsiveness to non-preferred stimuli[21], our data support the latter mechanism.

**Lack of specificity in V1 and subcortex**. Cortical and subcortical visual structures exhibited robust responses, though no selectivity for faces, even at the earliest dates. We collected data as early as 10 days of age, but measured strong visual responsiveness in cortex only after 4 weeks of age. At these earliest ages we did observe strong visual activations in the lateral geniculate nucleus (LGN) (Supplementary Fig. 5), which suggests that the lack of strong activations in cortex prior to 4 weeks was not due to a lack of activity-dependent vascular responses or failure of the monkeys to look at the stimuli. Our blood-volume measurements also showed a later emergence of both responsiveness in V1 and selectivity in IT than previous electrophysiological studies. Hubel and Wiesel[22] reported that V1 neurons in 3- and 4-week-old monkeys show not only visual responsiveness but also ordered sequences of orientation selectivity, and Kiorpes and Movshon[23] reported visual responsiveness and orientation selectivity in V1 as early as 1 week of age. Yet, we found strong fMRI responsiveness in the LGN and only weakly in V1 before 4 weeks of age. Furthermore, Rodman and colleagues reported the existence of face-selective neurons in IT of monkeys as young as 39 days old[18,19]. Is this discrepancy due to lack of sensitivity in fMRI, poor looking behaviour, or differences in what is measured by the two techniques? We do not think either lack of sensitivity or poor looking behaviour can be the explanation, given that LGN visual responsiveness was robust at 10 days; this LGN visual responsiveness indicates that the animal was looking at the stimulus, and that stimulus-induced blood-volume changes could be detected in the LGN. The blood-volume changes we measure may instead reflect population-level properties that were also manifest as smaller, slower, less reliable responses in the infant electrophysiological studies[18,19,22,23], as well as a smaller activity-dependent signal in both occipital cortex and IT using 2-deoxyglucose uptake (another metabolic measure) before 3 months of age[24,25].

We calculated mean beta coefficients for central V1, the LGN, the pulvinar and the amygdala in response to different image categories. Supplementary Fig. 6 shows longitudinal activations for the mean LGN and mean central V1 signals. There was no significant difference between any image categories, though responses in V1 were often higher for objects than for faces, consistent with the voxelwise maps in Fig. 2. Because it has been proposed that, during development, subcortical face mechanisms precede and drive the development of cortical face processing[26], we looked at the ventral pulvinar and amygdala, which have been shown in adults to respond to faces[27,28]. The ventral pulvinar showed significant activations to all three image categories, and the amygdala did not, but neither showed significant differential activation to any one category over any other at any point during development (Supplementary Fig. 7). Although we cannot say whether the lack of responsiveness in the amygdala is due to technical limitations or immaturity of the amygdala, this result, nevertheless, does not provide support for the hypothesis that either the amygdala or pulvinar precede and drive cortical face selectivity.

**Face-patch selectivity stabilizes around 6 months**. Once face patches emerged, selectivity for faces was stable and consistent. To show in a concise way the entire longitudinal pattern of the development of selectivity in IT, we calculated beta coefficients for the face and object-selective ROIs (Fig. 5; as above, ROIs were calculated from the data from monkey B3 on day 284 and from monkey B4 on day 276 thresholded at $P < 0.05$ FDR corrected and aligned onto the data for all other dates). When we looked at the (future) face and object patches in IT, visual responses were apparent in the face ROIs from the earliest sessions, with clear differences in category response magnitude emerging only later. Both anterior and middle face ROIs showed a significant preference for faces over objects and for faces over scrambled faces by 200 days of age in both monkeys. This is consistent with the contrast maps in Fig. 2 and the time courses in Fig. 4. Note that responses were calculated using data independent of the data used to define the ROIs, except for the single scan session for each monkey that was used to define the ROIs. Similarly, the object ROIs showed differential activations to objects compared to faces or scrambled faces also by 200 days of age in both monkeys. The responses to scrambled faces were intermediate.

The relative category selectivity (faces > objects in both face ROIs and objects > faces in the object ROI) was clear and stable by 200 days of age, but the response magnitudes were variable. This variability in response magnitude might reflect common factors, such as attentiveness, coil positioning or MION concentration that could vary from one scan session to another. We reasoned that V1 responsiveness should vary with all these factors and would represent a proxy for scan quality from session to session. We therefore normalized the ROI data by the V1 response, and in these normalized data the responsiveness in all patches showed a more stable and less variable plateau (Fig. 5, bottom row). In both face patches, in both monkeys, the V1 normalized response to faces increased up to around 200 days, then decreased slightly or remained flat. In contrast, the V1 normalized responses to objects in both face ROIs decreased over time (slopes significantly negative, see Fig. 5 legend). This decrease in responsiveness to what would become non-preferred stimuli supports the indication from the raw time courses in Fig. 4 that pruning contributes to the development of selectivity in face domains. Responses to both faces and objects decreased during development in the 'object' patches, but this may be because we did not use the actual preferred stimuli for these regions.

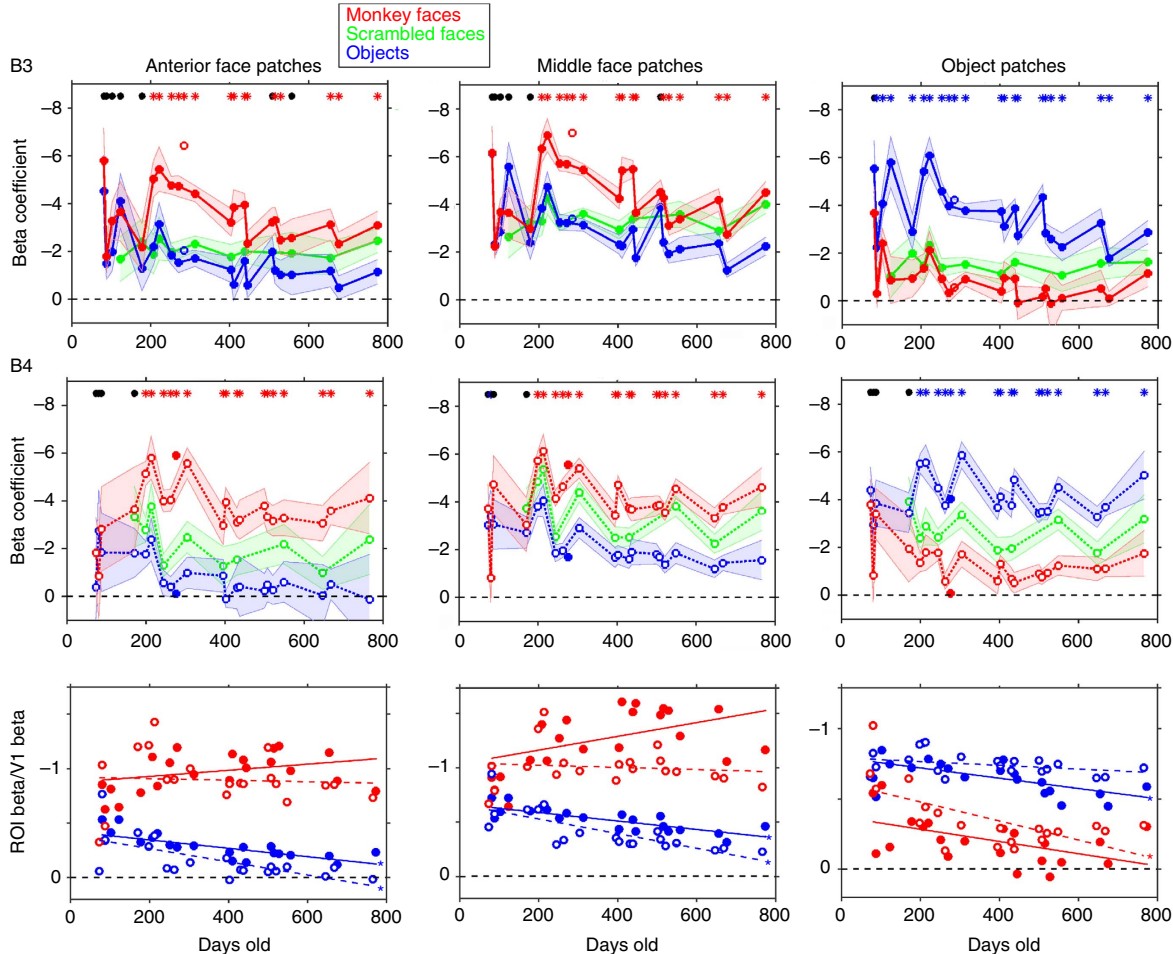

**Figure 5 | Development of category-selective responses in face and object ROIs in two individual monkeys for all the static-image sessions.** Responses to static-image blocks of faces (red), scrambled faces (green) and objects (blue) in the anterior-face ROI, the middle-face ROI, and in the object-selective ROI for monkey B3 (top row) and monkey B4 (middle row). Shading indicates ± s.e.m. Red asterisks at the top of each graph indicate sessions in which that ROI showed a significantly greater response to faces than to objects at $P < 0.05$; blue asterisks indicate sessions in which that ROI showed a significantly greater response to objects than to faces at $P < 0.05$; black dots indicate sessions in which there was no significant difference between faces and objects. Data from the scan sessions from which the ROIs were calculated are not connected by lines to the rest of the dates, and are indicated by open symbols for B3 and filled symbols for B4. (Bottom row) Responses by age in each ROI to faces (red) and objects (blue) divided by the responses in central V1 for that day to that image category in the same two monkeys. Lines indicate linear fit to each category data. B3 filled symbols, solid lines; B4 open symbols, dashed lines. Asterisks indicate slopes that were significantly less than zero at $P < 0.01$. Anterior face patch, slope of object responses (relative to y intercept at birth): B3, $m = -33\%/\text{year}$, $P = 5 \times 10-5$; B4, $m = -57\%/\text{year}$, $P = 6 \times 10-4$. Middle-face patch, slope of object responses: B3, $m = -21\%/\text{year}$, $P = 2 \times 10-5$; B4, $m = -37\%/\text{year}$, $P = 1 \times 10-4$. The slopes for face responses in the face patches were not significantly different from zero ($P > 0.05$). For both monkeys the slopes for the face responses in the face patches were significantly different from the slopes for object responses at $P < 0.01$ (assessed by a 3-way ANOVA with age, category, and face patch; interaction between age and category was significant at $F > 7.09$, $P < 0.01$; no other significant interactions, $P > 0.10$).

To ask how well the different IT domains could distinguish faces, scrambled faces and objects over development, we calculated a D-prime sensitivity index for each condition pair, for each ROI, and each scan session, for monkeys B3 and B4 (Fig. 6). D-primes were calculated with beta coefficients from a general linear model (GLM) in which the mean V1 signal was included as a nuisance variable to control for potential confounding factors discussed above (also see Methods).The ability to distinguish faces from objects increased from birth to between 200 and 300 days in all three ROIs.

**ROI-independent analysis confirms face-patch development.** We had chosen our non-face images to be as unlike faces as possible, to maximize the possibility of revealing any kind of selectivity

in IT as early as it might be present. On the basis of previous results in adolescents and adults, category-selective domains are correlated with a gradient of curvature/rectilinearity, with face-selective domains being correlated with preference for curvature[7,29]. We therefore chose to contrast monkey faces with rectilinear objects. In some scan sessions we also included blocks of scrambled faces, which share low-level shape, local structure and colour with faces, but do not share semantic category. Response magnitudes to scrambled faces were intermediate between responses to faces or objects in both face and object ROIs (Fig. 5). Since scrambled faces and objects are both artificial categories, but scrambled faces share low-level features (overall shape, local statistics, and colour) with faces, an interesting question is how the spatial distribution of scrambled-face responses compared to the spatial distribution of face and object responsiveness. We therefore used an unbiased

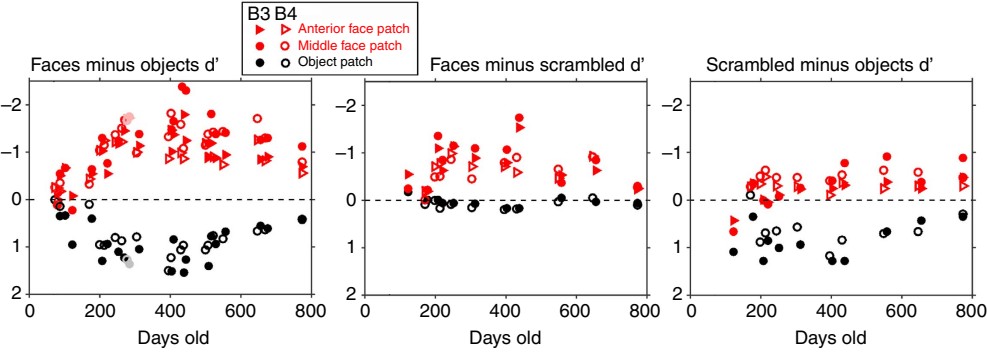

**Figure 6 | Development of category selectivity in face and object ROIs.** Plots show D-primes for distinguishing faces from objects (left), faces from scrambled faces (middle) and scrambled faces from objects (right) for the anterior face-patch cluster, middle face-patch cluster and object patches, as indicated. Data from the scan sessions from which the ROIs were calculated (B3 284 days; B4 276 days) are indicated by desaturated symbols in the first panel; we did not present scrambled faces in that session.

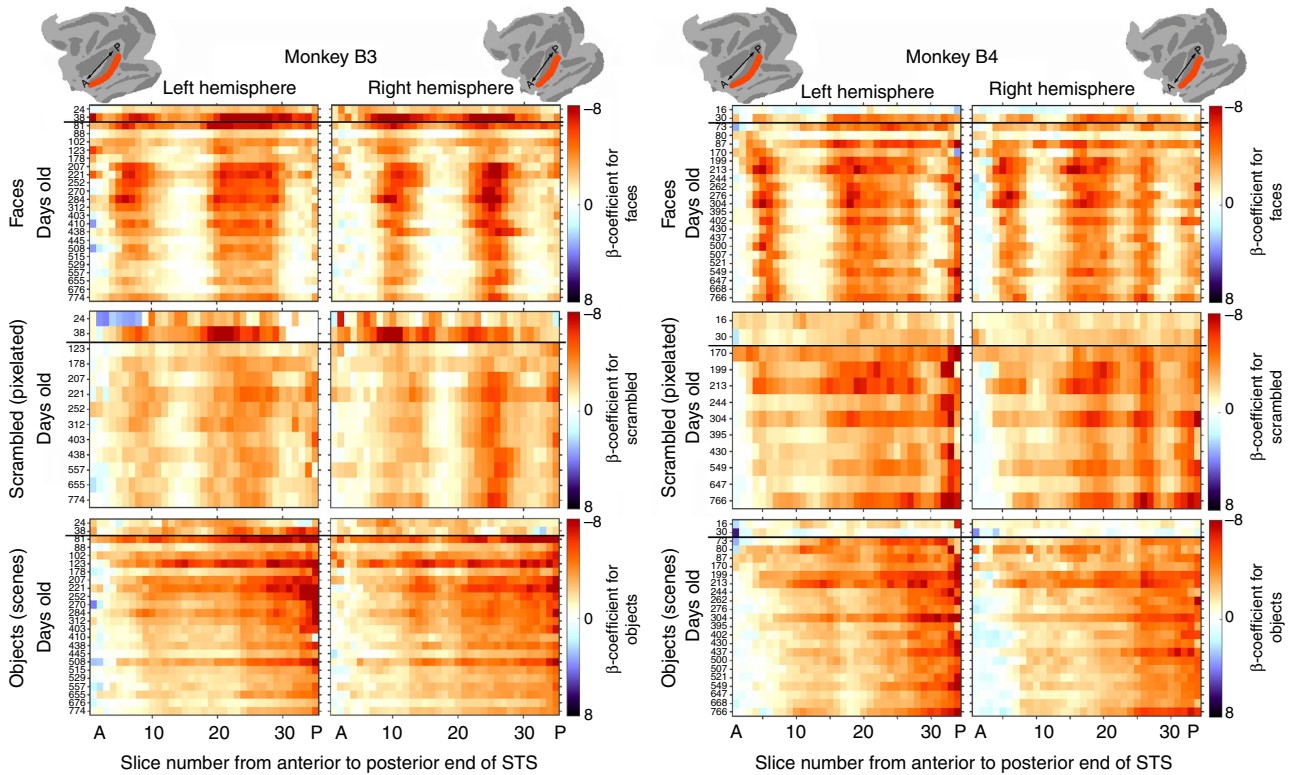

**Figure 7 | ROI-independent analysis of IT organization.** Signal change in response to faces/face movies (top row), scrambled faces/pixelated movies (middle row) and objects/scene movies (bottom row) as a function of age (vertical dimension) and distance along the STS from anterior to posterior (horizontal dimension). Black lines separate early movie data from later static-image data.

approach that condenses the spatial pattern but does not rely on the correspondence of ROIs from one age accurately corresponding to category-selective regions at another age. We drew an anatomical ROI that encompassed the entire anterior-to-posterior length of the lower lip of the STS (ROI diagrammed at the top of Fig. 7) and segmented this ROI into 35 slices perpendicular to the STS. Figure 7 shows the response in each slice at each age to each image category. The top row of matrices shows responses to faces. Larger responses (red) to faces are apparent as discrete peaks, the more anterior two peaks correspond to the face-patch clusters, and the activations at the posterior margin correspond to the location of the anterior border of V4. The locations of the face peaks were remarkably stable over time, even including several of the earliest

scans when the face and object patches were not statistically significant at $P < 0.05$.

**Evidence for a proto-map.** The pattern of activity elicited in IT by scrambled faces is clearly similar to the pattern for faces, and both are distinguishable from the pattern of responses to objects (Fig. 7). The middle row of matrices in Fig. 7 shows the signal change in each slice to blocks of scrambled faces; it shows that scrambled faces elicit responses in the same loci as did intact faces—two peaks in each hemisphere of monkey B3, one narrow and one broad region in monkey B4's left hemisphere, and three narrow peaks in monkey B4's right hemisphere. In contrast,

the responsiveness to objects (bottom matrices) exhibits a smoothly decreasing responsiveness from posterior to anterior in both monkeys. This pattern suggests an organization for shape, rather than semantic category. The similarity in location between the faces-minus-objects and scrambled-minus-objects activations (Supplementary Fig. 9) further indicates that low-level features common to faces and scrambled faces can selectively activate face domains. Lastly, activation maps for movies (Fig. 3) indicate that the early faces > scenes movies activations could be accounted for by low-level features, because the same loci were also activated by pixelated > scene movies.

As a further way to compare the similarity of face and scrambled-face activation patterns, we mapped outlines of the thresholds ($P < 0.05$ FDR corrected) for scrambled-faces-minus-objects onto the activation maps for faces-minus-objects (Supplementary Fig. 9). Although the scrambled > objects patches were weaker than the faces > objects patches, they were in the same locations, even as early as 153 days, which was the earliest that we saw scrambled > objects patches that reached a threshold of $P < 0.05$. The activity patterns and contrast maps further confirm that faces and scrambled faces differentially activate the same parts of the STS when compared to objects. Thus, an organization for low-level features common to faces and scrambled faces was present earlier than 200 days of age. Taken together, these data support the idea of a proto-organization selective for low-level features with a subsequent refinement of face selectivity.

## Discussion

We used blood-volume fMRI to monitor the emergence of the face-patch system in normally developing macaque monkeys. Infant macaques showed evidence for a proto-organization that preceded the face-patch system as early as 1 month of age. This organization underwent refinement of selectivity over the first year of development. Robust face > object selectivity was established by 200 days of age, though more subtle modifications continued to refine the organization up to about a year. If we use the 4:1 metric, 200 days would be equivalent to 2 years of age in a human[30], clearly an age later than face recognition abilities first emerge[31]. This raises the possibility that face patches are not the neural substrate for the face-processing abilities in early infancy.

Then what is the relationship between face recognition behaviour and face patches in IT? It has been reported that newborn humans[32] and infant monkeys with no face experience[33], preferentially look at faces, and that by 2–3 weeks of age monkeys respond socially to faces and differentiate between facial expressions[34,35]. Therefore face looking behaviour must precede the emergence of fMRI-detectable face-selective domains in IT. That is, fMRI-detectable face patches cannot be the neural substrate for early face looking behaviour. Morton and Johnson[26] proposed that subcortical structures drive face looking behaviour in early infancy, and this looking behaviour permits the maturation of the cortical face recognition system. However, we found no evidence for an early selectivity for faces in any subcortical structure.

If cortical face patches do not subserve early face looking behaviour, then the converse is that cluster formation may arise as a consequence of looking, consistent with our previous demonstration that extensive training can result in the formation of unnatural domains[7]. We cannot at this point resolve what the benefit is in adults of such clustering[6], or whether it is a byproduct of early learning mechanisms[36,37]. We previously found that intensive early experience can cause domain formation for unnatural image categories, and that these domains are

localized according to image features characteristic of each image set[7]. To explain this result we proposed the existence of a retinotopic proto-map in IT that carries with it a gradation in both spatial resolution and curvature selectivity. Our results here also support the idea of a shape-biased retinotopic proto-map that is refined by experience, in that we observe very early coarse selectivity for face-like stimuli, but this selectivity is refined during development, in particular by decreasing responsiveness to non-preferred stimuli. This decrease in responsiveness to non-preferred categories is consistent with findings of Cantlon et al.[38] in human children showing that the refinement of category selectivity in the fusiform gyrus also reflects decreasing responsiveness to non-preferred stimuli.

In movie sessions, we observed broad face selectivity as soon as strong visual responsiveness appeared in IT—as early as one month of age. The location of this early selectivity was similar to the location of the face-selective domains in the same animals at older ages. Therefore, one key finding of this study is that anatomical substrates for what will become the face-patch system must be present very early, if not at birth. Yet, this organization showed changes over the entire first year of development. What does this early proto-organization represent? We found early selectivity in future face patches for scrambled-minus-objects, neither of which is a natural image category. Therefore, the differential distribution of responsiveness to these image categories must be due to differences in their average image features or statistics and not their semantic category membership. A number of studies, using both fMRI[39–41] and physiology[42] have emphasized the importance of shape in driving categorical responsiveness; for example, face cells tend to respond to round non-face objects[43,44], and 'fish cells' to guitars[42]. In these studies on adults, the causal relationship between category selectivity and selectivity for shapes common to that category could not be determined. That is, given the different image statistics of different image categories[45], an apparent categorical organization could arise from a shape-based organization, or an apparent shape-biased organization could arise from extensive experience of the different image statistics of different categories. Here, we show that emerging category domains exhibit low-level shape or feature selectivity as early as they show visual responsiveness, indicating the existence of a very early, stable, proto-organization for shape, rather than a learned association between category and shape.

Do our data distinguish between the two major, seemingly incompatible, hypotheses for how face domains arise: an innate template for a biologically important semantic category or an innate proto-organization that turns out to distinguish some categories from others? The very early emergence of a stable stereotyped pattern of differential responsiveness supports an early innate organization. However, the selectivity increased over time, and was not absolute. The broadness of the selectivity is apparent in Fig. 7 and Supplementary Fig. 9, which show similar spatial organization for scrambled-face responsiveness and face selectivity. Further, it has been previously reported that even in adults face domains are also not absolutely face-selective, being broad enough to include other round objects and even curvy patterns[7,29,43]. The shared selectivity for faces and scrambled faces cannot be explained by an initial semantically defined face template that is broadened, by experiencing the image statistics of faces, to include other round/curvy things; rather the broad selectivity itself must be innate. Therefore, our data are consistent with either an innate categorical face template that is broad enough to include grapefruits, clocks and swirls, or an innate low-level proto-map that is differentially biased for roundish things and rectilinear objects. Distinguishing between the two models is beyond the scope of our data, but boils down to the question of

whether the kind of coarse selectivity we observe can be generated by an activity-dependent self-organizing system, or requires domain-specific constraints. Given the complexity that emerges as early as V1 in what is thought to be an activity-dependent self-organizing map[46,47], it is not inconceivable that this kind of selectivity could also emerge in a hierarchical self-organizing system. Our data further indicate that this functional organization is refined subsequent to the onset of visual experience, so, if the proto-organization is retinotopic[7,48], this refinement could be particularly affected by eye gaze patterns.

## Methods

**Monkeys.** fMRI studies were carried out on four experimentally naïve infant/juvenile *Macaca mulattas*, three male and one female, all born in our laboratory. Animals were housed under a 12-h light/dark cycle. All procedures conformed to USDA and NIH guidelines and were approved by the Harvard Medical School Institutional Animal Care and Use Committee. The monkeys were co-housed with their mothers in a room with other monkeys for the first 4–6 months, then co-housed with other monkeys, also in a room with other monkeys. Animal number per cage was within USDA guidelines by weight. For scanning they were alert (except when they fell asleep), and their heads were immobilized non-invasively using a foam-padded helmet with a bite bar. We used helmets of increasing size as the monkeys grew. For the youngest monkeys, we used a bite bar with a rubber nipple to hold the jaw and deliver formula; when they were around 3 months of age, we switched to a bite bar that delivered sweet juice. The monkeys were scanned in a primate chair that was modified to accommodate small monkeys in such a way that they were positioned upright when they were <2 months old, but positioned semi-upright, or in a sphynx position as they got larger. They were always positioned so that they could move their bodies and limbs freely, but their heads were restrained in a forward-looking position by the padded helmet. The monkeys were rewarded with formula or juice for looking at the screen. Gaze direction was monitored using an infrared eye tracker (ISCAN, Burlington, MA, USA). We used a combination of gaze direction data and responses in V1 to determine which scans to use for analysis. A concern with longitudinal functional imaging is ascertaining whether any observed changes are due to neurological development or to age-related changes in behaviour. In humans, excess motion in young subjects is an issue[3,8]. In our study, motion was less of a problem because we restrained the head effectively, but determining whether the monkey was alert and looking at the stimulus was a problem, especially in the youngest monkeys, who tended to become inattentive or fall asleep, and in whom accurate eye tracking was difficult because of their small eye size and poor corneal reflection[49].

**Stimuli.** The visual stimuli were projected onto a screen at the end of the scanner bore; the stimuli are shown in Fig. 1. We used both video clips and blocks of static images. To maximize the possibility of measuring any visual responsiveness, in the first two scan sessions for three of the monkeys, we used video clips. We discontinued using movies once the monkeys started looking at the static images during acclimation sessions (∼2 months) because static images present fewer confounds, such as differing motion characteristics. All the images covered 20° of visual field, and the category items in the images were at least 10° across. Thus the monkeys' acuity even at 1 month should have been high enough to discriminate these large images[48].

**Movies.** The video clips were each 30 s long, preceded by 10 s of grey screen and followed by 20 s of grey screen. Each video was presented in a separate scan. The movies were 20° in height and 25° wide. They showed (1) monkeys moving around and grooming each other, with prominently visible faces; (2) the same videos but with the faces pixelated out, or (3) videos of traffic scenes or time-lapse videos of natural scenes (flowers blooming, leaves waving, vistas panning). The motion in the faces and pixelated faces movies was equivalent, since the movies were identical except for the pixelated faces. The motion in the time-lapse nature and traffic clips was similar in speed to the movement in the monkey movies, but was more global than the face and pixelated-face movies that mainly had local motion like hands performing grooming.

**Static images.** For monkeys 2 months and older, we used blocks of static images. Each scan comprised blocks of each image category; each image subtended 20° × 20° of visual angle and was presented for 0.5 s; block length was 20 s, with 20 s of a neutral grey screen between image blocks. Blocks and images were presented in a counterbalanced order. For the first few static-image sessions, we used only two image categories: monkey faces and rectilinear objects, to ensure that we obtained enough repetitions to detect clear responses; in later sessions, when the monkeys would work longer, we added blocks of scrambled faces, and other stimulus categories. All images were centred on a pink-noise background and subtended a similar maximum dimension of 10°. All images were equated for

spatial frequency and luminance using the SHINE toolbox[50]. Each scrambled face was synthesized using the method of Portilla and Simoncelli[51] to match the statistics of one of the monkey-face images. This algorithm reproduces the local structure but not the global structure of the starting image. The resultant scrambled images were masked by a rotated outline of one of the monkey faces.

**Scanning.** Monkeys were scanned in a 3-T Tim Trio scanner with an AC88 gradient insert using 4-channel surface coils (custom made by Azma Maryam at the Martinos Imaging Center). Each scan session consisted of 10 or more functional scans. We used a repetition time (TR) of 2 s, echo time (TE) of 13ms, flip angle (α) of 72°, iPAT = 2, 1 mm isotropic voxels, matrix size 96 × 96 mm, 67 contiguous sagittal slices. To enhance contrast and measure blood volume directly[10,52], we injected 12 mg kg$^{-1}$ monocrystalline iron oxide nanoparticles (MION; Feraheme, AMAG Parmaceuticals, Cambridge, MA, USA) in the saphenous vein just before scanning. MION increases the signal-to-noise and inverts the signal[10]; for the readers' convenience we show the negative signal change as upwards in all our plots. By using this contrast agent, we were able to reliably obtain clear visual response signals in V1 for single scans, so we could use the V1 signal as a quality control, to eliminate blocks or entire scans when the monkey was inattentive or not looking at the stimulus.

**Data analysis.** Functional scan data were analysed using AFNI[53] and Matlab (Mathworks, Natick MA). Each scan session for each monkey was analysed separately. Potential 'spike' artifacts were removed using AFNI's 3dDespike. All images from each scan session were then motion corrected and aligned to a single reference time point for that session. Scans with movements >1 mm were not included in any further analysis. Data were spatially smoothed with a small kernel (1 mm full width half maximum (FWHM)) to increase signal-to-noise while preserving spatial specificity. Each scan was normalized to its mean. For each scan, signal quality was assessed within a ROI that encompassed the entire opercular surface of visual area V1 (which covered the central 6–7° of visual field, confirmed by retinotopic mapping at a later age). Square-wave functions matching the time course of the experimental design were convolved with a MION hemodynamic function[10]. Pearson correlation coefficients were calculated between the response functions and the average V1 response. We used both gaze direction data and average V1 responses to evaluate data quality. We deleted entire scans if the monkey was asleep, was not looking at the screen for at least 75% of the scan, or if the response in V1 did not fit the MION hemodynamic response convolved with the stimulus time-course with a correlation coefficient >0.5. We also censored individual blocks that showed spikes in the V1 signal or poor correlation with the convolved stimulus from the regression analysis. When individual blocks were censored, conditions were re-balanced by censoring blocks from the other stimulus conditions starting with the last scan, which typically had the poorest data quality and largest animal movement. Importantly, only V1 responses were used for evaluation; responses within the STS were not considered during this selection process. This scan and block selection was necessary for quality control in these young monkeys, and inclusion of bad scans would have given a false perspective on the responses in each session. The number of blocks of each category used for analysis for each date for each monkey is listed in Supplementary Table 1.

A multiple regression analysis (AFNI's 3dDeconvolve[53]) in the framework of a general linear model[54] was then performed for each scan session separately. Each stimulus condition was modelled with a MION-based hemodynamic response function[10]. Additional regressors that accounted for variance due to baseline shifts between time series, linear drifts, and head motion parameter estimates were also included in the regression model. For the data presented in Fig. 6, a separate GLM was estimated with an additional nuisance regressor consisting of the MION signal averaged over a region of interest in retinotopically-defined central V1. This additional V1-regression model was used to eliminate common factors, such as attentiveness, coil positioning, or MION concentration that could vary within and between scan sessions. Owing to the time-course normalization, resulting beta coefficients were scaled to reflect % signal change. Brain regions that responded more strongly to monkey faces, scrambled monkey faces, or familiar rectilinear objects were identified by contrasting presentation blocks of each of these image categories. Maps of beta coefficients were clustered (>10 adjacent voxels) and threshold at $P<0.05$ (FDR corrected). For early scans where no significant clusters for faces-minus-objects were found in the STS, the threshold was reduced to $P<0.25$ FDR corrected, as indicated in Fig. 2.

ROIs for the LGN, ventral pulvinar (below the brachium of the colliculus), and V1 were drawn on the anatomical template for B3 and B4 using retinotopic mapping data from the same monkeys at >2 years of age. ROIs for the middle face-patch cluster (consisting of ML and MF), for the anterior face-patch cluster (consisting of AL and AF) were calculated from faces-minus-objects data from monkey B3 at 285 days old and B4 at 276 days old using a criterion of $P<0.05$ (FDR corrected) for faces>objects; the object patches were calculated on the same data using the criterion of $P<0.05$ (FDR corrected) for objects>faces; the object patch was clipped posteriorly so that it included only object-selective regions within inferotemporal cortex anterior to V4. All ROIs were then projected onto the anatomy of each scan session using a two-step linear then nonlinear alignment (JIP Analysis Toolkit, written by Joseph B. Mandeville of the Martinos Imaging Center).

An STS-lower-lip ROI was drawn on the anatomical template of B4 day 284 and B5 day 276 (see top of Fig. 7 for illustration). This ROI included the entire lower bank and lip of the STS, from its anterior tip to the prelunate gyrus and V4 (identified by retinotopic mapping) posteriorly. Although the ROI was not constrained by functional data, it was chosen to optimally sample face and non-face-selective cortex, such that the anatomically defined strip would alternately traverse, from anterior to posterior: non-face-selective cortex to the most likely location of the anterior face patch, then to non-face cortex, then to the middle-face patch and finally to non-face cortex. We chose this ROI to compare the responsiveness of regions that would and would not become face patches before clear face selectivity emerged, without assuming anatomical correspondence between the youngest and the older brains. This STS-lower-lip ROI was then projected onto the anatomical templates for all the other sessions of each monkey. For each scan session, the STS-lower-lip ROIs were divided into 35 slices (perpendicular to the STS along the AP axis; slices were 1mm wide for the reference date, but were not necessarily the same width for other dates since we used the same number of slices for all dates despite changing brain size). Because the face patches are elongated in the mediolateral direction, and spaced along the AP direction, beta coefficients for all the voxels in each slice were averaged along ML and SI axes. The STS ROI did not optimally sample the object-selective regions, which were located more ventral; including them would have resulted in slices along the anterior–posterior axis that mixed both face- and object-selective regions at the likely locations of face patches.

The average time courses for each ROI (Fig. 4) were extracted from preprocessed data. Each time-course was baseline normalized to the average of three time points preceding each stimulus block onset, and detrended using a linear offset function for the single-block scans, or a second order polynomial for multi-block time courses. Stimulus blocks were averaged together to get an average signal change and confidence limits were calculated across all blocks for each stimulus condition.

D-primes (Fig. 6) were calculated from V1 normalized beta coefficients after removing the mean V1 response via regression in the GLM. Then, beta coefficients and critical $t$-values were used to calculate standard errors. These values were used to calculate a D-prime index using the following formula[55,56]:

$$d' = \frac{\mu_{\text{preferred}} - \mu_{\text{non-preferred}}}{\sqrt{\frac{\sigma^2_{\text{preferred}} + \sigma^2_{\text{non-preferred}}}{2}}}$$

where $\mu_{\text{preferred}}$ and $\mu_{\text{non-preferred}}$ are $-1$ (to invert the MION response) × the average responses to the preferred stimulus category (for example, faces) or the average responses to the non-preferred stimulus category (for example, objects); $\sigma_{\text{preferred}}$ and $\sigma_{\text{non-preferred}}$ are the s.d.

**Data availability.** Data will be available on request.

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

## Acknowledgements

This work was supported by NIH Grants RO1 EY 25670, P30 EY 12196 and F32 EY 24187. This research was carried out in part at the Athinoula A. Martinos Center for Biomedical Imaging at the Massachusetts General Hospital, using resources provided by the Center for Functional Neuroimaging Technologies, P41EB015896, a P41 Biotechnology Resource Grant supported by the National Institute of Biomedical Imaging and Bioengineering (NIBIB), National Institutes of Health and NIH Shared Instrumentation Grant S10RR021110. We thank A. Schapiro, Doris Tsao and M. Pinsk for helpful comments on the manuscript.

## Author contributions

All authors scanned monkeys; T.S. and P.F.S. trained monkeys; M.S.L. and M.J.A. analysed the data and wrote the paper.

## Additional information

**Competing interests:** The authors declare no competing financial interests.

**Publisher's note**: 

