## [Peer review file · Nature Communications]

Reviewers' comments:

Reviewer #1 (Remarks to the Author):

This is an outstanding study reporting unique and highly important data that informs our understanding of the development of cortical selectivity. There's no doubt that this will be a highly significant and influential paper. By scanning monkeys longitudinally over a protracted age range from very early in life, this study provides critical extensions to work from the same group and new insights about the time course of development.

That said, I think the manuscript can be substantially strengthened in several ways.

First, the manuscript comes across as quite disorganized. Related analyses are often split across different sections of the manuscript, there's much redundant repetition across different sections and the overall logic of the presentation order was not at all clear to me. For example the section on 'Subcortical and V1 responses' seems completely out of place and disrupts the flow of the other analyses. I highly recommend the authors rethink the ordering of the different sections, try to group related analyses together and provide some sort of 'roadmap' at the beginning of the results section so the reader knows where everything is heading. Ultimately, the authors present most of the analyses I wanted to see, but I had to work very hard to put everything together.

Second, I think the authors may be overselling their evidence for pruning as the explanation for the changes in selectivity. The best evidence is perhaps presented in Figure 5C, but the authors need to show not only that there is significant negative slope for the non-preferred stimuli, but that the slope measured is significantly different from the slope for the preferred stimuli. Currently, the authors do not even seem to calculate the slope for the preferred stimuli. If the authors can demonstrate a significant difference in the slopes that would strengthen their argument. Without it, I would recommend the authors be a little more cautious in pushing the pruning account.

Third, Figure 2 is comprehensive and gives a nice impression of the longitudinal data, but at the same time some of the data are occluded. I would recommend the authors produce a "movie" file of the maps in supplementary material that a reader could step through (at least for monkeys B3 and B4). I would also recommend the authors produce both thresholded and unthresholded versions.

Reviewer #2 (Remarks to the Author):

The authors longitudinally track the development of face responses in monkey cortex using fMRI, from about a month of age to several months. They find that discriminative responses become evident around 200 days and are stable subsequently. Some precursors to these responses are also evident at earlier ages. This study represents an impressive amount of work and the data are likely to be of some interest to researchers in the domain of face processing. However, their impact and interpretability are rather diminished by the lack of behavioral data and technique limitations.

On line 177, the authors state that “We collected data as early as 1 week of age, but we saw clear visual responses in cortex only after 4 weeks of age.” This is surprising and seems to arise from a limitation of the recording technique rather than an intrinsic visual non-responsivity in the cortex. There have, of course, been studies showing robust single unit responses in infant monkeys much younger than 4 weeks of age (e.g. Kiorpes and Movshon, 2004, *Vis. Neurosc.* 21(6)). The lack of visual responses until 4 weeks in this study raises questions about the robustness and sensitivity of the recording methodology, and hence, the interpretations that may be derived from the data.

The more significant shortcoming of this work, I believe, is the lack of a behavioral assay that can be used to interpret the neuroimaging results. In other words, how capable were the monkeys in telling apart faces from non-faces at different points in their developmental timeline? And, how does the development of behavior correlate with the development of neural responses? Given their early emerging social responses, one would presume that they could accomplish this discrimination well before 200 days of age. How then can one interpret the onset of a significant distinction in cortical responses so late? There are at least two possibilities, both not very encouraging in terms of the significance of the results reported here: 1. The responses in the localized face ROIs are not the causal substrate for behavioral performance. Some other set of responses, perhaps spatially distributed across multiple areas, might be performing this role, but are not revealed because the analysis adopted here is not designed to pick up on spatially distributed patterns of activity. 2. If the chosen ROIs are indeed the causal substrates for face classification, then perhaps the BOLD measurements collected are not robust enough for revealing neural selectivity early in timeline.

A minor point: On line 186, the authors state: “These traces show, first, that the responses in an ROI for central V1 were significant and robust at every session. This validates that, for each block included in the averages, the monkeys were indeed looking at the stimuli”

Perhaps I am misunderstanding the authors' claim, but it appears to me that the presence of V1 responses only indicate that the monkeys were looking (i.e. they had their eyes open). How can the authors be sure that the monkeys were looking at the screen rather than somewhere else?

In summary, although I have some concerns about the study, I am nevertheless impressed with the authors' addressing a logistically hard problem. The data are not as interpretable as one would have ideally wished them to be, but they may still prove to be of interest to researchers of object and face processing.

Reviewers' comments:

Reviewer #1 (Remarks to the Author):

This is an outstanding study reporting unique and highly important data that informs our understanding of the development of cortical selectivity. There's no doubt that this will be a highly significant and influential paper. By scanning monkeys longitudinally over a protracted age range from very early in life, this study provides critical extensions to work from the same group and new insights about the time course of development.

That said, I think the manuscript can be substantially strengthened in several ways.

First, the manuscript comes across as quite disorganized. Related analyses are often split across different sections of the manuscript, there's much redundant repetition across different sections and the overall logic of the presentation order was not at all clear to me. For example the section on 'Subcortical and V1 responses' seems completely out of place and disrupts the flow of the other analyses. I highly recommend the authors rethink the ordering of the different sections, try to group related analyses together and provide some sort of 'roadmap' at the beginning of the results section so the reader knows where everything is heading. Ultimately, the authors present most of the analyses I wanted to see, but I had to work very hard to put everything together.

We agree with the reviewer that the organization of the paper may be confusing. We had a difficult time figuring the best order in which to present our results. It still seems most logical to us to first describe the emergence of the face patches, and then, in order to demonstrate that our failure to find early face patches was not due to monkeys not looking or lack of sensitivity, to bring up V1 and subcortical structures. If we showed the V1 and subcortical structures first, we couldn't compare those responses to face patches, because we wouldn't have yet defined the face patches. Also, the V1 responses provide an anchor by which to define the lack of specificity of the early (in time) responses, and help answer the question of whether those early responses were specific or not. To better orient the reader, we now lay out the logic of the order of our results in the introduction, and we have changed the sub-section titles to explicitly state what the point is of each section. Below is how we revised the introduction to summarize the logical flow:

"Here we monitored the emergence and refinement of the most well-studied IT domain, face patches, longitudinally from 1 month to more than 2 years of age using fMRI of alert macaques over their normal development. We contrasted responses to static images of faces with responses to images of rectilinear objects and found that face patches emerged around 200 days of age, and were remarkably stable after that. In the youngest, least attentive, animals we used movie clips that contrasted intact animals with the same animals with pixelated faces or with inanimate dynamic scenes. These movies revealed some selectivity for faces compared to scenes as early as 1 month of age. In order to figure out whether the very early selectivity to the movies represented face selectivity per se or some low-level selectivity to shapes that include faces, we compared responses to faces and objects to responses to scrambled faces, which share local image features with faces, and we compared the selectivity of the responses in IT to responses in V1 and in subcortical structures. We determined that the very early face selectivity was weaker than the selectivity that emerged by 200 days of age. The early broad selectivity for faces was in the same location as the later, stronger, selectivity, indicating the existence of an early proto-organization."

Here is how we changed the subsection titles: "Subcortical and V1 Responses → Lack of specificity in V1 and subcortex; Quantitative ROI Analysis → Face patch selectivity stabilizes around 6 months; ROI-Independent Analysis → ROI-Independent Analysis confirms face patch development. We hope these changes make the logical flow clearer, but are open to further suggestions."

Second, I think the authors may be overselling their evidence for pruning as the explanation for the changes in selectivity. The best evidence is perhaps presented in Figure 5C, but the authors need to show not only that there is significant negative slope for the non-preferred stimuli, but that the slope measured is significantly different from the slope for the preferred stimuli. Currently, the authors do not even seem to calculate the slope for the preferred stimuli. If the authors can demonstrate a significant difference in the slopes that would strengthen their argument. Without it, I would recommend the authors be a little more cautious in pushing the pruning account.

We now show the slopes for all the image categories in Figure 5C and indicate which are significantly different from zero. We found a significant difference in slopes only in the face patches; we have added this to the text. We did

not find significantly different slopes in the object patches, but we don't know what the actual preferred category is for the 'object' patches; we know only that it is not faces. But this is not our only evidence that reduced responses to non-preferred categories contributes to selectivity: decreasing response to objects in face patches are also apparent from the raw timecourses in Fig. 4.

Third, Figure 2 is comprehensive and gives a nice impression of the longitudinal data, but at the same time some of the data are occluded. I would recommend the authors produce a "movie" file of the maps in supplementary material that a reader could step through (at least for monkeys B3 and B4). I would also recommend the authors produce both thresholded and unthresholded versions.

We now include a supplementary figure showing non-overlapping maps of the development of the face patches in monkeys B3&B4, showing the entire flattened cortex for each session. We hope the reviewer appreciates that nothing was being hidden by overlapping the maps, rather we wanted to show the entire developmental timecourse in as compact a way as possible. Unthresholded maps are uninformative because the significant voxels are indistinguishable from the insignificant data, which is just noise. We did lower the threshold below statistical significance for those maps that showed nothing in the STS, and we have made this clearer by stating: "Note that where the age is blue in Fig. 2 there were no significant faces>object activations at all in the STS, so the threshold was lowered to $p < 0.25$ in order to reveal any statistically insignificant organization that might be present."

Reviewer #2 (Remarks to the Author):

The authors longitudinally track the development of face responses in monkey cortex using fMRI, from about a month of age to several months. They find that discriminative responses become evident around 200 days and are stable subsequently. Some precursors to these responses are also evident at earlier ages. This study represents an impressive amount of work and the data are likely to be of some interest to researchers in the domain of face processing. However, their impact and interpretability are rather diminished by the lack of behavioral data and technique limitations. We have added a paragraph to discuss previously published behavioral data (see below), and we show in Figure 4 that our technique is sensitive enough to reveal responsivity prior to the emergence of selectivity (see below).

On line 177, the authors state that "We collected data as early as 1 week of age, but we saw clear visual responses in cortex only after 4 weeks of age." This is surprising and seems to arise from a limitation of the recording technique rather than an intrinsic visual non-responsivity in the cortex. There have, of course, been studies showing robust single unit responses in infant monkeys much younger than 4 weeks of age (e.g. Kiorpes and Movshon, 2004, *Vis. Neurosc.* 21(6)). The lack of visual responses until 4 weeks in this study raises questions about the robustness and sensitivity of the recording methodology, and hence, the interpretations that may be derived from the data.

We have added a paragraph to acknowledge and address the discrepancy between our blood volume measurements and the single neuron neurophysiology studies of Hubel and Wiesel as well as Rodman & Gross and Kiorpes & Movshon. We think it is unwarranted to conclude from this discrepancy that measuring blood volume is necessarily insensitive, given that

1. We did find significant visual responses in the LGN and to a lesser extent in peripheral V1, just not in IT, in our earliest fMRI scans (at 10-11 days). We have added a supplementary figure Fig. S5 to document this.
2. Blood-volume fMRI may measure population-level features that are different from the electrical activity of single neurons, and that something may well be immature before 1 month of age.

Here is what we have added:

"Cortical and subcortical visual structures exhibited robust responses, though no selectivity for faces, even at the earliest dates. We collected data as early as 10 days of age, but measured strong visual responsiveness in cortex only after 4 weeks of age. At these earliest ages we did observe strong visual activations in the LGN (Fig. S5), which suggests that the lack of strong activations in cortex prior to 4 weeks was not due to lack of activity-dependent vascular responses or failure of the monkeys to look at the stimuli. Our blood volume measurements also showed a later emergence of both responsiveness in V1 and selectivity in IT than previous electrophysiological studies. Hubel and Wiesel reported that V1 neurons in 3 and 4 week old monkeys show not only visual responsiveness but also ordered sequences of orientation

selectivity¹, and Kiorpes and Movshon reported visual responsiveness and orientation selectivity in V1 as early as 1 week of age². Yet we found strong fMRI responsiveness in the LGN and only weakly in V1 before 4 weeks of age. Furthermore, Rodman and colleagues reported the existence of face-selective neurons in IT of monkeys as young as 39 days old^{3,4}. Is this discrepancy due to lack of sensitivity in fMRI, poor looking behavior, or differences in what is measured by the two techniques? We do not think either lack of sensitivity or poor looking behavior can be the explanation, given that LGN visual responsiveness was robust at 10 days; this LGN visual responsiveness indicates that the animal was looking at the stimulus, and that stimulus-induced blood volume changes could be detected in the LGN. The blood volume changes we measure may instead reflect population-level properties that were also manifest as smaller, slower, less reliable responses in the infant electrophysiological studies¹⁻⁴, as well as a much smaller activity-dependent signal in both occipital cortex and IT using 2-deoxyglucose uptake (another metabolic measure) before 3 months of age^{5,6}.”

The more significant shortcoming of this work, I believe, is the lack of a behavioral assay that can be used to interpret the neuroimaging results. In other words, how capable were the monkeys in telling apart faces from non-faces at different points in their developmental timeline? And, how does the development of behavior correlate with the development of neural responses? Given their early emerging social responses, one would presume that they could accomplish this discrimination well before 200 days of age. How then can one interpret the onset of a significant distinction in cortical responses so late? There are at least two possibilities, both not very encouraging in terms of the significance of the results reported here: 1. The responses in the localized face ROIs are not the causal substrate for behavioral performance. Some other set of responses, perhaps spatially distributed across multiple areas, might be performing this role, but are not revealed because the analysis adopted here is not designed to pick up on spatially distributed patterns of activity. 2. If the chosen ROIs are indeed the causal substrates for face classification, then perhaps the BOLD measurements collected are not robust enough for revealing neural selectivity early in timeline. Regarding possibility #1: We agree that the face patches may not be the neural substrate for early face-looking behavior in primates. Defining when face-looking behavior emerges, when face discrimination appears and matures, compared to when face patches emerge, should answer that question. We are monitoring the looking behavior and face discrimination of infant monkeys, and, indeed, we do find that normal monkeys look at faces prior to face patches emerging. The data we are collecting on infant face-looking behavior go far beyond the scope of the submitted manuscript, so would not yet be appropriate to include here. Indeed, given that several studies have already been published showing very early face-looking behavior in infant monkeys, this does imply that face patches are not the causal substrate for at least some face looking behavior. This in turn suggests the converse: that extensive exposure to faces may instead be causal for the formation of face patches. We now discuss the results from these studies with our imaging data in the following, which we have added to the discussion:

“Then what is the relationship between face recognition behavior and face patches in IT? It has been reported that newborn humans⁷ and infant monkeys with no face experience⁸, preferentially look at faces, and that by 2-3 weeks of age monkeys respond socially to faces and differentiate between facial expressions^{9,10}. Therefore face looking behavior must precede the emergence of fMRI-detectable face-selective domains in IT. That is, fMRI-detectable face patches cannot be the neural substrate for early face looking behavior. Morton and Johnson¹¹ proposed that subcortical structures drive face looking behavior in early infancy, and this looking behavior permits the maturation of the cortical face recognition system. We found no evidence for an early selectivity for faces in any subcortical structure. If cortical face patches do not subserve early face looking behavior, then the converse is that cluster formation may arise as a consequence of looking, consistent with our previous demonstration that extensive training can result in the formation of unnatural domains¹².”

Regarding possibility #2: We provide evidence that our measurements were robust enough for revealing selectivity early in development: First, we did not measure BOLD signal, as stated by the reviewer, but rather blood volume using an iron contrast agent, which is much more sensitive than BOLD, possibly because BOLD pits the deoxygenation of hemoglobin against increased blood flow. Second, we showed that our signal was strong and reliable (Figure 4), even before the signal was selective, so lack of sensitivity cannot be the explanation for the early lack of selectivity.

A minor point: On line 186, the authors state: “These traces show, first, that the responses in an ROI for central V1 were significant and robust at every session. This validates that, for each block included in the averages, the monkeys were indeed looking at the stimuli” Perhaps I am misunderstanding the authors’ claim, but it appears to me that the presence

of V1 responses only indicate that the monkeys were looking (i.e. they had their eyes open). How can the authors be sure that the monkeys were looking at the screen rather than somewhere else?

The signal in V1 was time-locked to the presentation of images on the screen. Nothing else the monkeys could have been looking at changed at that frequency, since the non-image blocks had the same luminance as did the image blocks. Also, as stated, we monitored the monkey's eyes during scans to verify that they were indeed looking towards the screen.

In summary, although I have some concerns about the study, I am nevertheless impressed with the authors' addressing a logistically hard problem. The data are not as interpretable as one would have ideally wished them to be, but they may still prove to be of interest to researchers of object and face processing.

1. Wiesel, T.N. & Hubel, D.H. Ordered arrangement of orientation columns in monkeys lacking visual experience. *J Comp Neurol* **158**, 307-318 (1974)
2. Kiorpes, L. & Movshon, J.A. Neural Limitations on Visual Development in Primates. in *The Visual Neurosciences* (eds. Chalupa, L. & Werner, J.S.) (MIT Press, Cambridge, MA, 2004).
3. Rodman, H.R., Scalaidhe, S.P. & Gross, C.G. Response properties of neurons in temporal cortical visual areas of infant monkeys. *J Neurophysiol* **70**, 1115-1136 (1993)
4. Rodman, H.R., Skelly, J.P. & Gross, C.G. Stimulus selectivity and state dependence of activity in inferior temporal cortex of infant monkeys. *Proc Natl Acad Sci U S A* **88**, 7572-7575 (1991)
5. Bachevalier, J., Hagger, C. & Mishkin, M. Functional maturation of the occipitotemporal pathway in infant rhesus monkeys. in *Brain work and mental activity* (eds. Lassen NA, ME, R. & L, F.) 231-242 (Munksgaard, Copenhagen, 1991).
6. Distler, C., Bachevalier, J., Kennedy, C., Mishkin, M. & Ungerleider, L.G. Functional development of the corticocortical pathway for motion analysis in the macaque monkey: A 14C-2-deoxyglucose study. *Cereb Cortex* **6**, 184-195 (1996)
7. Johnson, M.H., Dziurawiec, S., Ellis, H. & Morton, J. Newborns' preferential tracking of face-like stimuli and its subsequent decline. *Cognition* **40**, 1-19 (1991)
8. Sugita, Y. Face perception in monkeys reared with no exposure to faces. *Proc Natl Acad Sci U S A* **105**, 394-398 (2008)2224224.
9. Kenney, M.D., Mason, W.A. & Hill, S.D. Effects of age, objects, and visual experience on affective responses of rhesus monkeys to strangers. *Developmental Psychology* **15**, 176-184 (1979)
10. Mendelson, M.J., Haith, M.M. & Goldman-Rakic, P.S. Face scanning and responsiveness to social cues in infant rhesus monkeys. *Developmental Psychology* **18**, 222-228 (1982)
11. Morton, J. & Johnson, M.H. CONSPEC and CONLERN: a two-process theory of infant face recognition. *Psychol Rev* **98**, 164-181 (1991)
12. Srihasam, K., Vincent, J.L. & Livingstone, M.S. Novel domain formation reveals proto-architecture in inferotemporal cortex. *Nat Neurosci* **17**, 1776-1783 (2014)4241119.

REVIEWERS' COMMENTS:

Reviewer #1 (Remarks to the Author):

The authors have fully addressed my concerns and I congratulate them on an impressive and important piece of work.

Reviewer #2 (Remarks to the Author):

I like the clarifications and revisions the authors have made in the manuscript. While the changes do not directly address the issue of how the neuro-imaging results relate to behavior, I feel that it would be unfair to hold up this paper on that count. I think that the developmental progressions in neural responses reported here will be interesting in their own right to many researchers. I believe the manuscript as it now stands will be well-received by the community and hence have no hesitation in recommending it for publication.